



# Gaussian mixture model for extreme wind turbulence estimation

Xiaodong Zhang, Anand Natarajan

Technical University of Denmark, Department of Wind Energy, Frederiksborgvej 399, 4000 Roskilde, Denmark

**Correspondence:** Xiaodong Zhang (xiazang@dtu.dk)

**Abstract.** Uncertainty quantification is a necessary step in wind turbine design due to the random nature of the environmental loads, through which the uncertainty of structural loads and responses under specific situations can be quantified. Specifically, wind turbulence has a significant impact on the extreme and fatigue design envelope of the wind turbine. The wind parameters (mean and standard deviation of 10-minute wind speed) are usually not independent, and it will lead to biased results for structural reliability or uncertainty quantification assuming the wind parameters are independent. A proper probabilistic model should be established to model the correlation among wind parameters. Compared to univariate distributions, theoretical multivariate distributions are limited and not flexible enough to model the wind parameters from different sites or direction sectors. Copula-based models are used often for correlation description, but existing parametric copulas may not model the correlation among wind parameters well due to limitations of the copula structures. The Gaussian mixture model is widely applied for density estimation and clustering in many domains, but limited studies were conducted in wind energy and few used it for density estimation of wind parameters. In this paper, the Gaussian mixture model is used to model the joint distribution of mean and standard deviation of 10-minute wind speed, which is calculated from 15 years of wind measurement time series data. As a comparison, the Nataf transformation (Gaussian copula) and Gumbel copula are compared with the Gaussian mixture model in terms of the estimated marginal distributions and conditional distributions. The Gaussian mixture model is then adopted to estimate the extreme wind turbulence, which could be taken as an input to design loads used in the ultimate design limit state of turbine structures. The wind turbulence associated with a 50-year return period computed from the Gaussian mixture model is compared with what is utilized in the design of wind turbines as given in the IEC 61400-1.

## 1 Introduction

Wind turbulence is characterized by the turbulence kinetic energy, its dissipation rate, and the length scale. This is modeled using three-dimensional anisotropic spectra that captures the auto-correlation and cross-correlation of the spatio-temporal wind speed variation such as through the Mann model (Mann, 1994). Such model assumes wind turbulence is a Gaussian process, whereby several frequencies of wind velocity variations may occur resulting in different wind velocities distributed as a function of time and space. Usually, the wind turbulence for wind turbine design is specified over a 10-minute time window and the stochastic process is assumed to be stationary. The occurrences of extreme turbulence can then be categorized based on its return period. In wind turbine design, the wind turbulence with a 50-year return period is used in ultimate limit state analysis (IEC, 2019).





Many uncertainties exist in the evaluation of the design loads of wind turbine components. The IEC 61400-1 standard lists several load cases of the relevance of ultimate limit state analysis, wherein the load cases under normal operation usually require a partial safety factor (PSF) of 1.35 applied to the characteristic loads. Such PSFs are determined by quantifying
the uncertainties in the load evaluation (Sørensen and Toft, 2014) and the underlying distributions of the relevant inputs. An important load-case towards determining ultimate design loads is the Design Load Case (DLC) 1.3, in which the turbine is under normal operation under 50-year extreme wind turbulence. While relationships to evaluate the extreme turbulence level are provided in the IEC 61400-1, there has been much debate on its quantification; with the edition-3 of the IEC 61400-1 specifying a lognormal distribution for turbulence and edition-4 specifying it as a Weibull distribution. Several studies (Dimitrov et al.,
2017; Abdallah et al., 2016) have proposed different models for extreme wind turbulence based on site measurements and a large uncertainty can be seen in determining the long-term behavior of wind turbulence. Mathematically, an issue with the modelling of wind turbulence has been that the IEC 61400-1 standard and literature has mainly focused on the probability distribution of wind speed standard deviation ($\sigma_u$) conditional on the mean wind speed ($u$), whereas it is required that the joint distribution of $\sigma_u$ and $u$ is properly modeled.

A proper description of the joint distribution of stochastic variables required for loads prediction is important for structural reliability analysis and uncertainty propagation. A joint distribution model could be used for modelling multivariate random variables and generating random samples. Theoretical bivariate distributions are limited and not flexible enough. (Monahan, 2018) model the joint probability distribution of wind speeds at different locations using bivariate Rice distribution and bivariate Weibull distribution. Using marginal distributions and copula to model the multivariate distributions is feasible, but the marginal
distributions should be flexible enough to represent the wind inflow under varying environmental conditions, and the tail of the fitted distribution should be well representative of the actual inflow behavior. The copula structures should also be flexible enough to model different correlation structures. It is not clear as to which copula model (Abdallah, 2015) to choose to determine the joint distribution, given marginal distributions.

It is a fundamental requirement for both the correlated random variables and the independent variables that the dependency
of the random variables should be well modelled in all regions of the distribution, including the tail. For modelling extreme turbulence accurately, the tail of the joint probability distribution of $\sigma_u$ and $u$, must be accurately represented to small exceedance probabilities of the order of $10^{-7}$. The Gaussian mixture model (GMM) is a flexible model which can perform density estimation on multivariate data with different marginal distributions and correlations. GMM is widely applied to different fields of study, e.g., speech and audio processing (Reynolds and Rose, 1995), image classification (Permuter et al., 2003),
density estimation of microarray data in bioinformatics (Steinhoff et al., 2003), cancer classification (Prabakaran et al., 2019) and finance (Miyazaki et al., 2014). GMM is less commonly applied in wind energy, (Srbinovski et al., 2021) used GMM for modelling the site-specific wind turbine power curves, (Chang et al., 2017) used GMM based neural network for short-term wind power forecast, (Cui et al., 2018) used GMM for fitting the probability distribution of wind power ramping features. (Li et al., 2020) used GMM for electrical loads forecast. For wind parameters modelling, (Wahbah et al., 2018) used univariate
GMM for wind speed probability density estimation, where the joint distribution of wind speed with other parameters was not





investigated. Few published literature uses GMM for density estimation of wind inflow parameters and GMM has not been used for modelling the joint distribution of mean wind speed and standard deviation.

In this paper, a GMM is used for modelling the joint distribution of wind parameters, i.e., 10-minute $u$ and $\sigma_u$. The GMM is firstly used for density estimation of a random sample from theoretical bivariate $t$ distribution. Then it is used for modelling the wind parameters from both offshore and onshore sectors. The GMM is benchmarked to the measurement data by comparing the marginal distributions and the conditional distributions. The 50-year turbulence is also computed from the GMM model. For the wind parameters from the offshore sector, Gaussian copula (Nataf transformation) and Gumbel copula are also compared.

## 2 Gaussian mixture model

The GMM (McLachlan and Peel, 2000) is a mixture of several weighted Gaussian distributions and has been used for cluster analysis (Janouek et al., 2015) and density estimation (Steinhoff et al., 2003). The GMM could be used for hard clustering and soft clustering of data. For hard clustering, each observation is assigned to the component returning the highest posterior probability, where each observation is assigned to exactly one cluster. Soft clustering, as opposed to hard clustering, assigns each observation to more than one cluster and each observation is assigned a responsibility (relative density). In terms of density estimation, The GMM is useful for multivariate distribution representations with multiple modes, but this does not prevent it from also being used for single mode distributions. The GMM is a linear combination of multivariate Gaussian distribution components, where each component is defined by its mean and covariance. The probability distribution function (pdf) of a $d$-dimensional multivariate Gaussian is

$$\mathcal{N}\left(\mathbf{x}|\mu,\textstyle\sum\right) = \frac{1}{\sqrt{|\sum|(2\pi)^d}} \exp\left(-\frac{1}{2}(\mathbf{x}-\mu)\textstyle\sum^{-1}(\mathbf{x}-\mu)^{\mathrm{T}}\right) \tag{1}$$

where $\mu$ is the 1-by-$d$ mean vectors, and $\sum$ is the $d$-by-$d$ covariance matrix. The pdf of GMM is

$$p(\mathbf{x}) = \sum_{j=1}^{k} \pi_j \mathcal{N}\left(\mathbf{x}|\mu_j,\textstyle\sum_j\right) \tag{2}$$

where $k$ is the number of components, which is a hyper parameter, and $\pi_j$ is the component coefficient (weight) and follows

$$\sum_{j=1}^{k} \pi_j = 1 \quad 0 \leq \pi_j \leq 1 \tag{3}$$

Some information criteria are proposed in the literature (Akaike, 1998; Schwarz, 1978) to determine $k$, but further research is needed to properly apply them when the sample size is too large. To use GMM for density estimation and also for random sample generation, the model parameters $\{\pi_j, \mu_j, \sum_j, j = 1, 2, ..., k\}$ should be estimated from the data sample $\{\mathbf{x}_n, n = 1, 2, ..., N\}$, where $N$ is the sample size. The model parameters are estimated by the following steps:

1. Assign the $N$ observations to the $k$ clusters using the $k$-means clustering algorithm. Compute $\mu_j$, $\sum_j$ and $\pi_j$ from the observations within each cluster.





$k$-means clustering assigns $N$ observations to $k$ clusters, which are defined by the centroids. Each data point $x_n$ with the closest centroid is assigned to the corresponding cluster. The centroids are recalculated and the data points are reassigned until the clusters do not change or the maximum iteration number is met. This is a hard clustering, and within each component, the $\mu_j$ and $\sum_j$ are calculated, and the $\pi_j$ is calculated as the number of data points in the current cluster divided by $N$.

2. Expectation-Maximization (EM) algorithm

The model parameters $\{\pi_j, \mu_j, \sum_j, j = 1, 2, ..., k\}$ are found by an iterative EM algorithm (Dempster et al., 1977) to have a maximum likelihood estimation.

(a) E step

Evaluate the responsibilities using the current model parameters. The responsibility $\gamma_j(\mathbf{x}_n)$ is the probability that component $j$ takes for explaining the observation $\mathbf{x}_n$, which is calculated as:

$$\gamma_j(\mathbf{x}_n) = \frac{\pi_j \mathcal{N}\left(\mathbf{x}_n | \mu_j, \sum_j\right)}{\sum_{i=1}^{k} \pi_i \mathcal{N}\left(\mathbf{x}_n | \mu_i, \sum_i\right)} \tag{4}$$

(b) M step

Update the model parameters using the responsibilities from E step. The mean for component $j$ is calculated as:

$$\mu_j = \frac{\sum_{n=1}^{N} \gamma_j(\mathbf{x}_n)\mathbf{x}_n}{\sum_{n=1}^{N} \gamma_j(\mathbf{x}_n)} \tag{5}$$

The covariance for component $j$ is calculated as:

$$\Sigma_j = \frac{\sum_{n=1}^{N} \gamma_j(\mathbf{x}_n)(\mathbf{x}_n - \mu_j)(\mathbf{x}_n - \mu_j)^T}{\sum_{n=1}^{N} \gamma_j(\mathbf{x}_n)} \tag{6}$$

and the $j$ component coefficient is calculated as:

$$\pi_j = \frac{1}{N} \sum_{n=1}^{N} \gamma_j(\mathbf{x}_n) \tag{7}$$

3. Repeat step 2 until the model parameters converge or the maximum number of iterations is met.

# 3   Results

A joint probability distribution of the 10-minute mean wind speed and turbulence is developed by fitting specific probability distributions and correlation functions to multi-year wind measurement data. The traditional method of using copulas to develop non-Gaussian joint-distributions is initially attempted.





### 3.1 Wind measurements

The wind measurements from the Høvsøre Test Centre for Large Wind Turbines in western Denmark (Dimitrov et al. (2017),

Hannesdóttir et al. (2019)) are used in this study. The 10-minute high-frequency time series of three-dimensional wind velocities at a height of 100 m is selected. The period of measurements is from 1 January, 2005 to 1 January, 2020, i.e. 15 years of measurement data (Hannesdóttir et al., 2019). Each 10-minute time series is used to calculate the $u$ component mean wind speed $u$, and is linearly detrended to calculate the standard deviation $\sigma_u$. The wind parameters from the offshore sector ($225°$ to $315°$) and onshore sector($150°$ to $180°$ and $45°$ to $135°$) are studied here. Outliers and potentially missing data elements

are omitted. The sample size is about $2.43 \times 10^5$ for the offshore sector, $4.09 \times 10^4$ for the onshore sector ($150°$ to $180°$), and $1.41 \times 10^5$ for the onshore sector ($45°$ to $135°$).

The joint distribution of random variables could be described by the univariate marginal distribution functions and a copula. A copula is a multivariate cumulative distribution function, where the marginal distribution follows uniform distribution on the interval [0,1]. Copulas are used for modelling the dependency among the random variables. The copula function models the

dependency structure of the random variables. Several families of copulas have been proposed in the literature, e.g., Gaussian copula (Nataf transformation (Xiao, 2014)), Archimedean copulas (Bouyé et al., 2011).

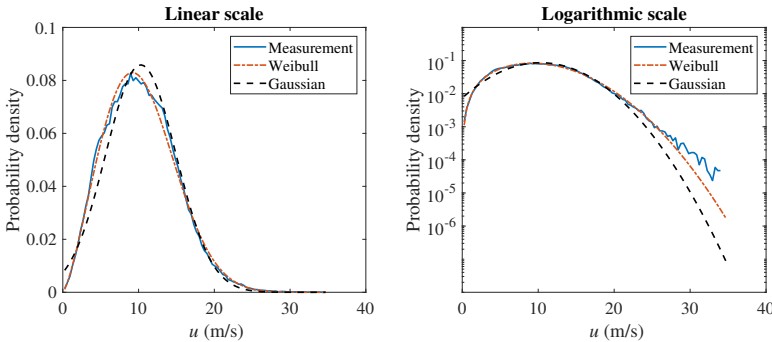

**Figure 1.** Marginal distribution of $u$ with Weibull fitting

The marginal distributions to be used are to be defined and the correlation between the variables is modelled by the copula structure. Here, a Weibull distribution is used for modelling the marginal distribution of $u$, where the scale parameter is 11.61 and the shape parameter is 2.35. The plots are shown in Fig. 1. The lognormal distribution is used for modelling the marginal

distribution of $\sigma_u$, where the mean and standard deviation of logarithmic values are -0.61 and 0.52. The plots are shown in Fig. 2. Both the linear and logarithmic scales are plotted, where the main body pdf and tail distribution could be compared. It could be seen that Weibull and lognormal distributions are fairly good fits for the $u$ and $\sigma_u$ respectively. The univariate Gaussian distribution is also used here to fit the distribution of $u$ and $\sigma_u$, but it is not a proper fit, which also indicates that the multivariate Gaussian distribution is not a good candidate for modelling the joint distribution of the wind parameters. The Nataf

transformation (Xiao, 2014) and Gumbel copula are used here to model the joint distribution of $u$ and $\sigma_u$ and generate random samples. The generated random sample is shown in Fig. 3, where the left figure is the scatter plot of the measurement data, the





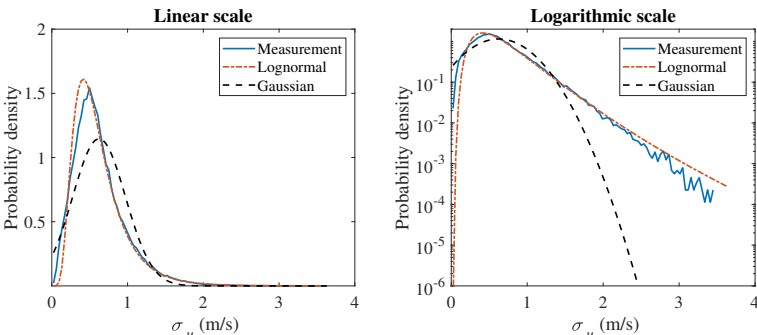

**Figure 2.** Marginal distribution of $\sigma_u$ with lognormal fitting

middle figure is the Nataf transformation generated sample, and the right figure is the Gumbel copula generated sample. The Nataf transformation and Gumbel copula generated samples have the same sample size as the measurement data. They have the same fitted marginal distributions but different copula structures, as is demonstrated in Fig. 3.

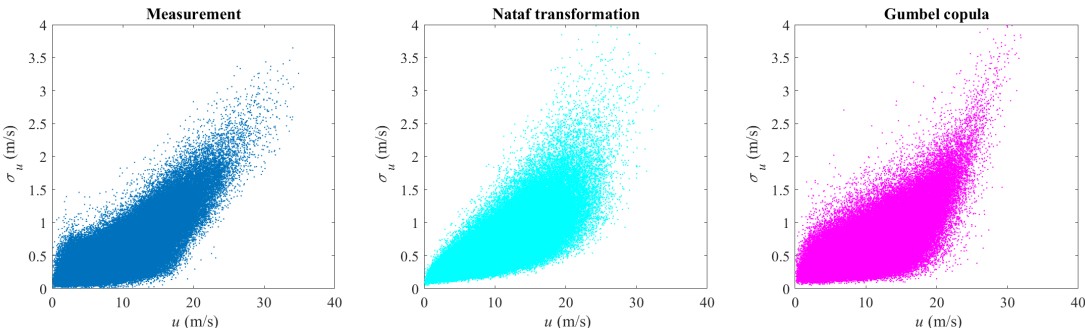

**Figure 3.** Nataf transformation and Gumbel copula random samples for offshore sector

The different copula structures lead to different conditional distributions. The Nataf transformation and Gumbel copula estimated probability of exceedance of $\sigma_u$ conditional on $u$ are shown in Fig. 4 and Fig. 5 respectively. Only the distributions $u \geq 16$ m/s are plotted as they are close to the tail and affect the 50-year turbulence estimation most. with As seen in Fig. 4, the probabilities of exceedance of $\sigma_u$ conditional on $u$ deviate from the measurement data significantly. Using a Gumbel copula as is shown in Fig. 5, even though there is a reasonable agreement when $u$ ranges from 16 m/s to 20 m/s, a larger discrepancy arises for higher mean wind speeds. The differences in the conditional distribution between the copula-estimated and measurement data indicate that using copula could lead to a biased 50-year turbulence estimation and large model uncertainty for DLC 1.3 simulations.

Even though other copula structures are available, they are not flexible enough to represent the joint distribution of $u$ and $\sigma_u$ from different measurement sites or even the same site for different wind direction sectors. The correct copula to use to generate the joint distribution of $u$ and $\sigma_u$ for tail estimation requires further research. However, instead of fitting the joint



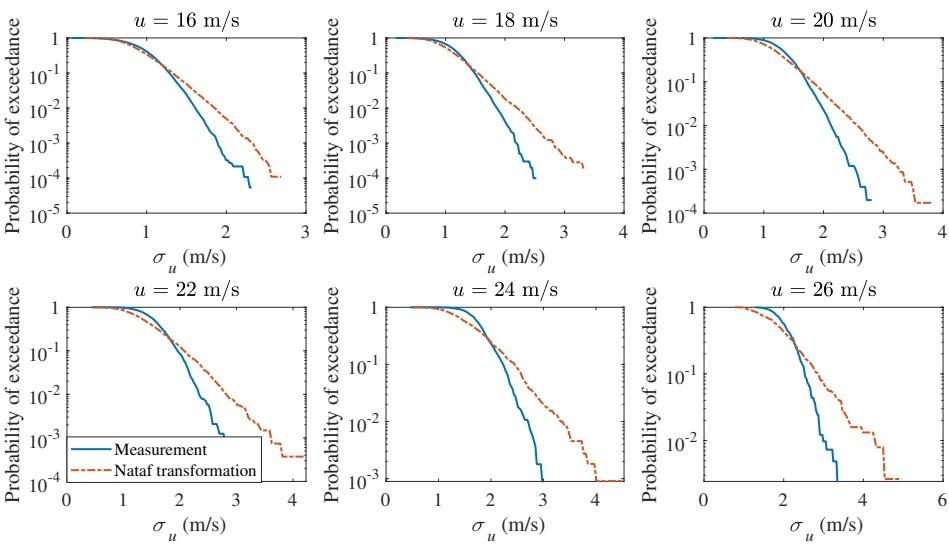

**Figure 4.** Nataf transformation probability of exceedance of $\sigma_u$ conditional on $u$

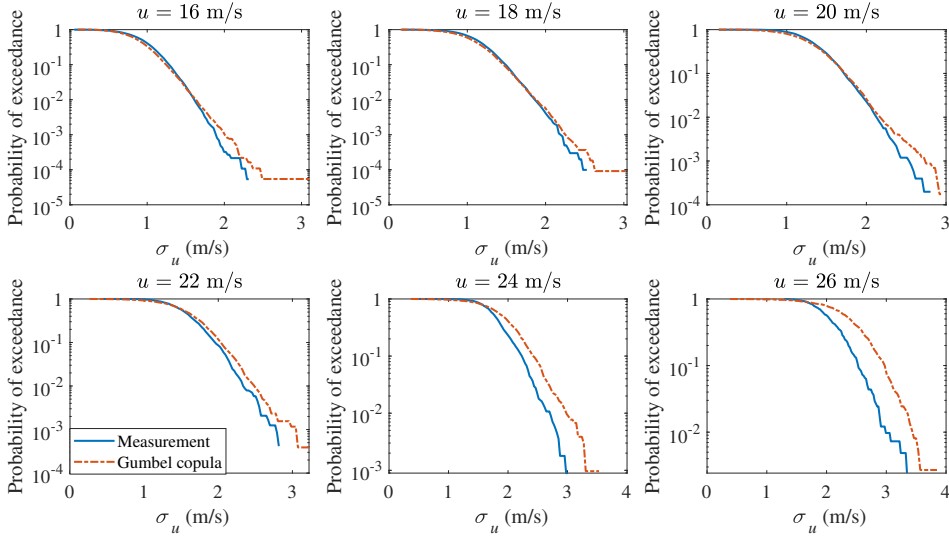

**Figure 5.** Gumbel copula probability of exceedance of $\sigma_u$ conditional on $u$

distribution using copula methods, a multivariate distribution is another option. To perform density estimation on univariate random variables, many theoretical probability distributions are available, e.g., normal, Weibull, lognormal, Rayleigh distribution, and the methods in (Zhang et al., 2020; Low, 2013), etc. On the other hand, fewer probability distributions are available for





multivariate density estimations. This creates a similar limitation of copula models, i.e., theoretical multivariate distributions
are limited and not flexible enough to model the $u$ and $\sigma_u$ measurements that possess different correlation structures.

The GMM on the other hand is quite flexible since a number of Gaussian distributions with corresponding weights could
be used to estimate the probability densities for multivariate variables and generate correlated samples. The GMM is applied
herein with a theoretical $t$ distribution to test whether it could be flexible enough to model existing non-Gaussian theoretical
distributions. It is then utilized to perform density estimation, random sample generation of the $u$ and $\sigma_u$ from different wind
direction sectors.

### 3.2 Multivariate $t$ distribution

The multivariate $t$ distribution is selected here to test the flexibility of GMM and also demonstrate the procedure of using GMM
for density estimation. The pdf of the $d$-dimensional multivariate Student's $t$ distribution is

$$f\left(\mathbf{x}, \textstyle\sum, v\right) = \frac{1}{|\sum|^{1/2}} \frac{1}{\sqrt{(v\pi)^d}} \frac{\Gamma\left((v+d)/2\right)}{\Gamma(v/2)} \left(1 + \frac{\mathbf{x}^{'}\sum^{-1}\mathbf{x}}{v}\right) \tag{8}$$

where $\sum$ is a correlation matrix with a correlation coefficient 0.6, and $v = 5$ is the degrees of freedom. The multivariate
Student's $t$ distribution generalizes the univariate Student's $t$ distribution, and its marginal distributions all have univariate
Student's $t$ distribution. The marginal distributions of multivariate Student's $t$ distribution have fatter tails than the normal
distribution. A random sample with size $10^5$ is generated from the bivariate $t$ distribution, and GMM is used to fit the bivariate
$t$ distribution.

**Table 1.** Initial GMM parameters

| Component number (i) | 1 | 2 | 3 | 4 |
|---|---|---|---|---|
| $\mu_i$ | $\begin{bmatrix} -1.972 & -2.037 \end{bmatrix}$ | $\begin{bmatrix} 1.978 & 1.966 \end{bmatrix}$ | $\begin{bmatrix} -0.552 & -0.522 \end{bmatrix}$ | $\begin{bmatrix} 0.521 & 0.511 \end{bmatrix}$ |
| $\sum_i$ | $\begin{bmatrix} 1.242 & 0.067 \\ 0.067 & 1.273 \end{bmatrix}$ | $\begin{bmatrix} 1.237 & 0.128 \\ 0.128 & 1.128 \end{bmatrix}$ | $\begin{bmatrix} 0.396 & -0.155 \\ -0.155 & 0.396 \end{bmatrix}$ | $\begin{bmatrix} 0.384 & -0.155 \\ -0.155 & 0.396 \end{bmatrix}$ |
| $\pi_i$ | 0.107 | 0.111 | 0.388 | 0.395 |

The number of components $k$ is set to four. The $k$-means clustering algorithm is used to cluster the data points into $k = 4$
components. The mean, covariance, and the component coefficient (sample size at each component divided by the total sample
size) calculated from each component are taken as initial parameters for GMM, which are shown in Table. 1. The four clusters
are plotted in Fig. 6, where the means are plotted in circles.

Following the procedure of EM algorithm, the model parameters are estimated, which is shown in Table 2. Fig. 7 shows
the random sample from $t$ distribution and the GMM, and Fig. 8 shows the corresponding contour plot. The random sample





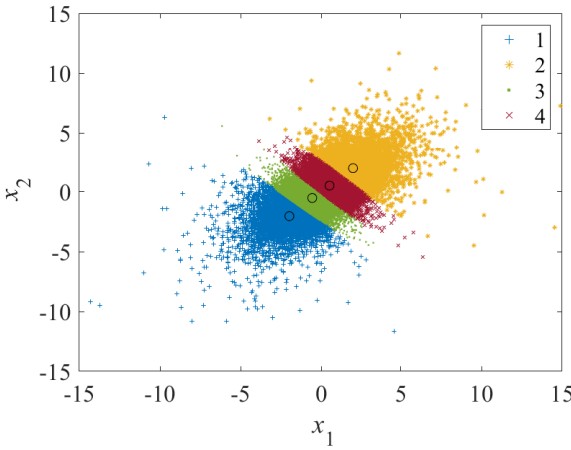

**Figure 6.** $k$-means clustering of $t$ distribution sample

from GMM has a similar correlation structure with the theoretical $t$ distribution. For probability densities higher than $10^{-5}$, the GMM agrees well with the theoretical $t$ distribution, for lower densities, $10^{-6}$ and $3.8 \times 10^{-7}$ (corresponding to 50 year return period for 10-minute measurement data), there is some deviation, which is due to the sample size and sample variation. As the densities estimated by the GMM is based on a random sample with sample size $10^5$ drawn from the theoretical $t$ distribution.

**Table 2.** Final GMM parameters

| Component number (i) | 1 | 2 | 3 | 4 |
|---|---|---|---|---|
| $\mu_i$ | $\begin{bmatrix} -1.80 & -0.736 \end{bmatrix}$ | $\begin{bmatrix} 0.016 & 0.001 \end{bmatrix}$ | $\begin{bmatrix} -0.011 & -0.004 \end{bmatrix}$ | $\begin{bmatrix} 0.014 & 0.015 \end{bmatrix}$ |
| $\sum_i$ | $\begin{bmatrix} 24.655 & 11.076 \\ 11.076 & 21.447 \end{bmatrix}$ | $\begin{bmatrix} 4.794 & 2.937 \\ 2.937 & 4.900 \end{bmatrix}$ | $\begin{bmatrix} 1.505 & 0.891 \\ 0.891 & 1.508 \end{bmatrix}$ | $\begin{bmatrix} 0.586 & 0.354 \\ 0.354 & 0.580 \end{bmatrix}$ |
| $\pi_i$ | 0.004 | 0.119 | 0.504 | 0.373 |

Note that the hyperparameter $k$ is set to four instead of being estimated, the GMM pdf converges when $k > 4$. Larger $k$ does not necessarily compensate the density estimation performance of GMM with a bit more computational effort.



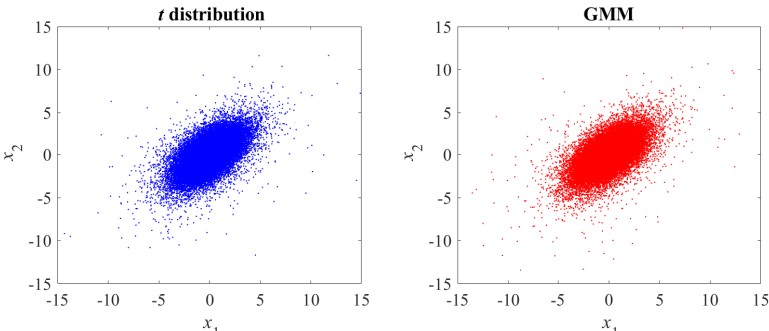

**Figure 7.** Random sample from $t$ distribution and GMM

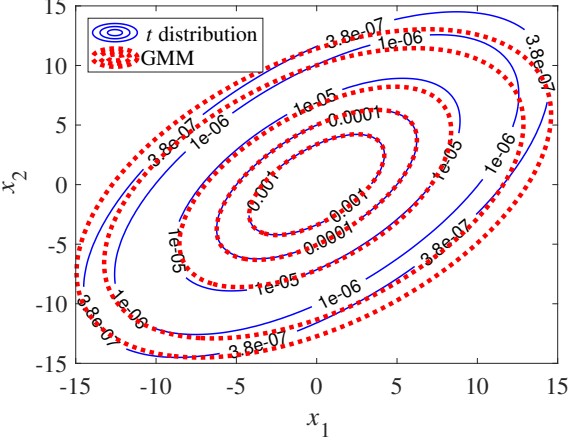

**Figure 8.** Contour plot of $t$ distribution and GMM





### 3.3 GMM based estimation of wind parameters for the offshore sector

It is important to model the joint distribution of wind parameters, which could be used for uncertainty quantification, structural optimization, and reliability analysis of wind turbines. The joint distribution should have a small estimation error for a realistic

50-year turbulence estimation. For the copula examples in section 3.1, the marginal distributions are estimated well, but not the correlation structure, which leads to the inaccuracy of the conditional distribution. If the focus is on the marginal distribution of $u$ and the distribution of $\sigma_u$ conditional on $u$, then the marginal distribution of $\sigma_u$ might be subject to estimation errors. Using GMM does not have the same limitation, as a good joint distribution estimation will estimate both marginal distributions and correlation structures with small estimation errors. Both the marginal distributions and conditional distribution estimated from

GMM are examined here.

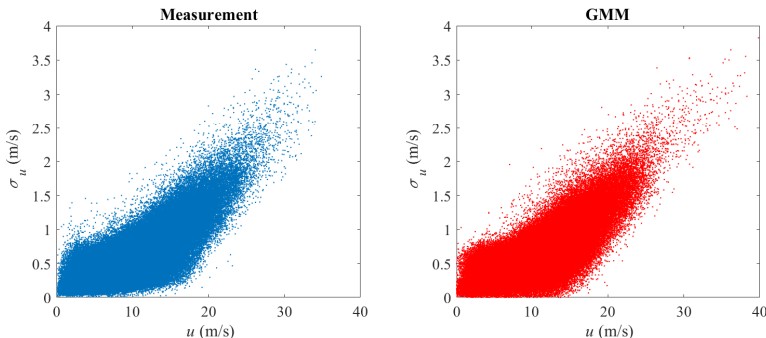

**Figure 9.** Measurement data and GMM random sample for offshore sector

The GMM is adopted here to model the joint distribution of $u$ and $\sigma_u$. The measurement data and the GMM random samples are shown in Fig. 9, where the correlation structure of the measurement data is well captured. The marginal distribution of $u$ is shown in Fig. 10 and the marginal distribution of $\sigma_u$ is shown in Fig. 11. Compared to Figs. 1 and 2, the marginal distributions from GMM has smaller difference with the measurement data at both the main body pdf and the tails. The univariate Gaussian

distribution is not a good fit for either of the marginal distribution, but GMM is a good fit as its marginal distribution is a linear combination of univariate Gaussian distributions, which is more flexible compared to a single Gaussian distribution. For marginal distribution estimation, which theoretical to choose remains a problem, especially when the sample size is small and the tail might exhibit different shapes. GMM does not have the trouble of selecting distributions for marginal distribution estimation.



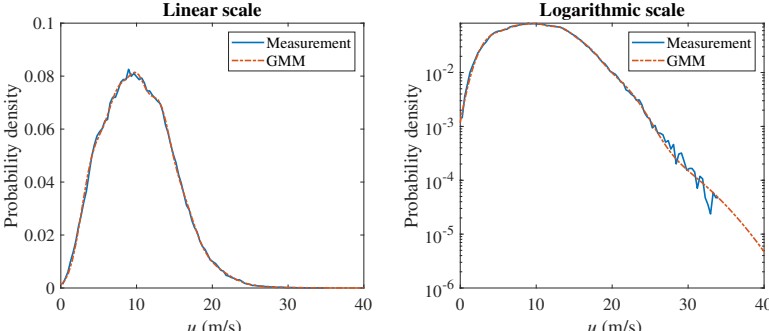

**Figure 10.** Marginal distribution of $u$ for offshore sector

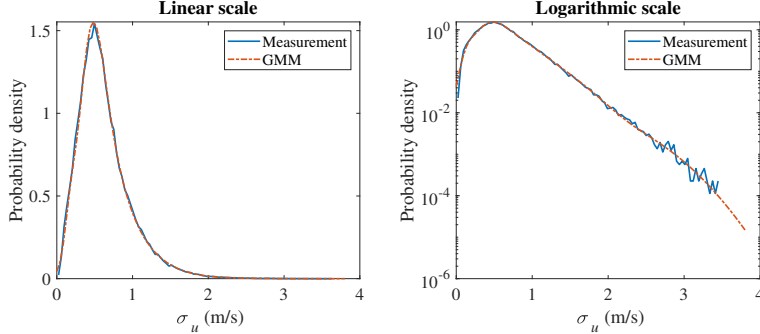

**Figure 11.** Marginal distribution of $\sigma_u$ for offshore sector

The probability distribution of $\sigma_u$ conditional on $u$ is plotted in Fig. 12. The probability of exceedance of $\sigma_u$ conditional on $u$ is plotted in Fig. 13. Both the main body pdf and the probability of exceedance from GMM agree quite well with the measurement data, for the bins when $u$ =26 m/s, the tail of the measurement data is not accurate due to the small sample size (412).





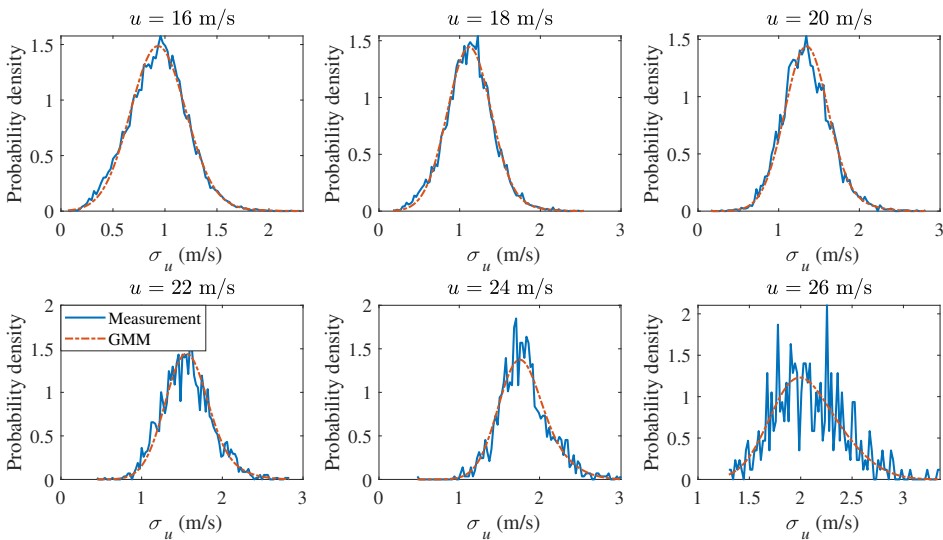

**Figure 12.** GMM probability distribution of $\sigma_u$ conditional on $u$ for offshore sector

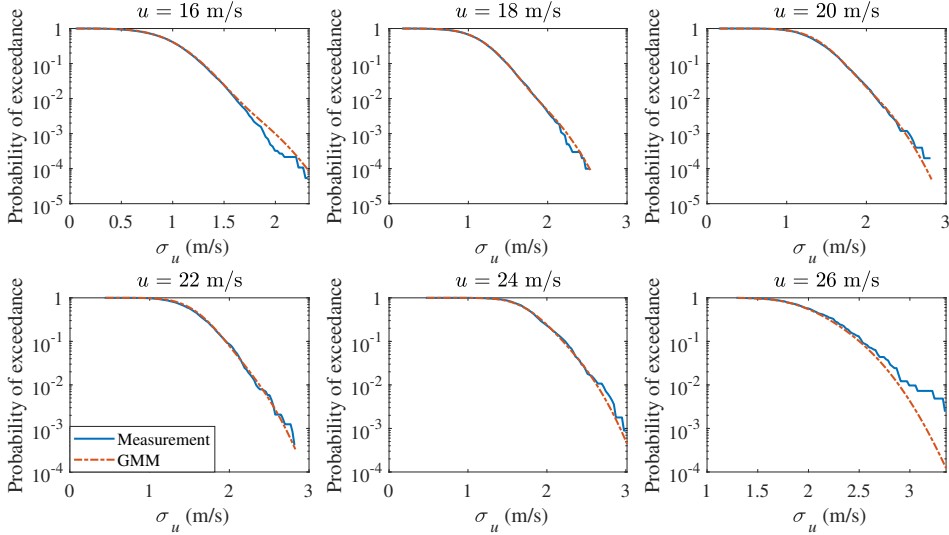

**Figure 13.** GMM probability of exceedance of $\sigma_u$ conditional on $u$ for offshore sector





The 10-minute turbulence level with a return period of 50 years is shown in Fig. 14. The 50-year turbulence levels have
probability density contour with a value of $3.805 \times 10^{-7}$. The contour labelled IEC (blue '+') uses a reference turbulence
intensity $I_{ref} = 0.12$ (corresponding to wind turbine class C) as input to perform IFORM analysis (IEC, 2005), where $u$ is
modeled by Weibull distribution and the probability distribution of $\sigma_u$ conditional on $u$ is modeled by lognormal distribution
(IEC, 2005). The IEC (data) (yellow '+') is the same as the IEC (blue '+') except that $I_{ref} = 0.057$, which is calculated as the
expected value of turbulence intensity at a mean wind speed of 15 m/s from the measurement data (IEC, 2005). The contour
labelled IEC (data) has lower values than the contour labelled IEC, since $I_{ref}$ is smaller (0.057 vs 0.12). The 50-year contour
estimated using the GMM is realistic as it has a similar shape to the scatter plot of the measurement data and bounds the data
points. The marginal distributions agree well with the measurement data (as are shown in Figs. 10 and 11), the conditional
distributions are validated in Figs. 12 and 13. The IEC contour happens to be aligned with the GMM, but the IEC 61400-1 does
not prescribe a joint probability distribution or the marginal distribution for $\sigma_u$. As the $I_{ref}$ used is much larger than obtained
through the measurement data (0.12 vs 0.057), it could be inferred that the use of a lognormal distribution conditional on the
mean wind speed or the empirical formulas in (IEC, 2005) might not be accurate. The fourth edition of the IEC 61400-1 (IEC,
2019) does not increase the accuracy with the Weibull distribution for turbulence conditional on the mean wind speed, as the
50-year turbulence level is still unchanged.

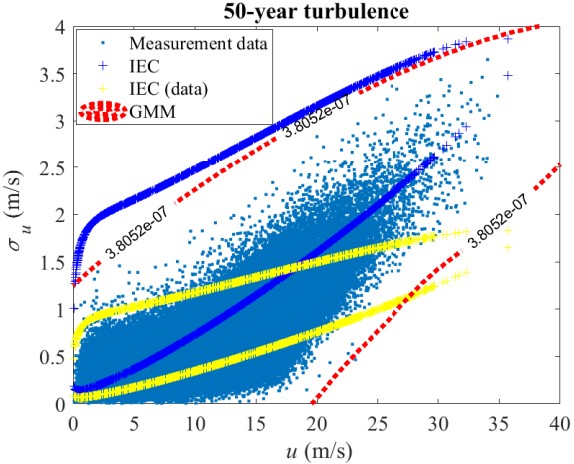

**Figure 14.** GMM and IEC 50-year turbulence estimation for offshore sector





### 3.4 GMM based estimation of wind parameters for the onshore sector

It is worth investigating the applicability of GMM to other wind direction sectors, where the wind parameters have different correlation structures due to different terrains. The wind velocity and turbulence in the onshore section (150° to 180°) is modelled using the GMM. The measurement data and a random sample from GMM are shown in Fig. 15, where the correlation structure is different from the offshore sector in Fig. 9. The marginal distribution of $u$ is shown in Fig. 16 and the marginal distribution of $\sigma_u$ is shown in Fig. 17. Negligible differences could be seen from the comparison of the main body pdfs and

the tails. The probability distribution of $\sigma_u$ on $u$ is plotted in Fig. 18. The probability of exceedance of $\sigma_u$ conditional on $u$ is plotted in Fig. 19. Note that the sample size is smaller than the offshore sector ($4.09 \times 10^4$ vs $2.43 \times 10^5$), so the tail distribution of the onshore measurement data has lesser accuracy as compared to the offshore sector, but still performs better than using the method of copulas. The 50-year turbulence contour is shown in Fig. 20, where the left figure shows the 50-year turbulence estimated from the measurement data from the sector with direction from 150° to 180°, and the right figure is from the sector

with direction from 45° to 135°. A slightly larger 50-year contour is estimated from the 45° to 135° sector. Figures 18, 19, and 20 show that the GMM is indeed flexible and can be used to model wind speed and turbulence for different wind conditions, albeit for flat terrains.

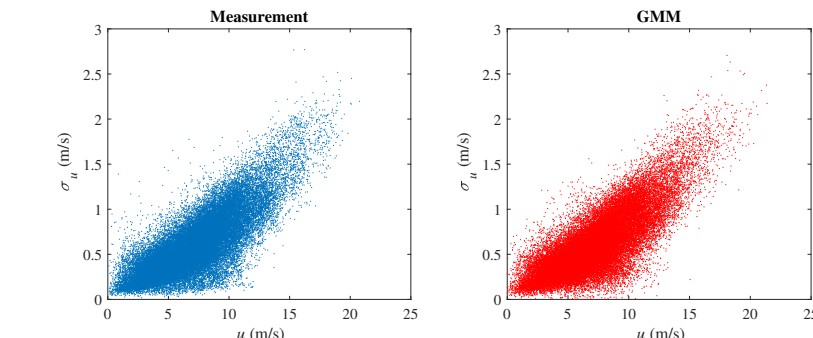

**Figure 15.** Measurement data and GMM random sample for onshore sector

     Note that $k$ is set to eight for all the wind parameters joint distribution estimation using GMM. More components are needed compared to the theoretical $t$ distribution as the correlation structure between the $u$ and $\sigma_u$ is more complex.



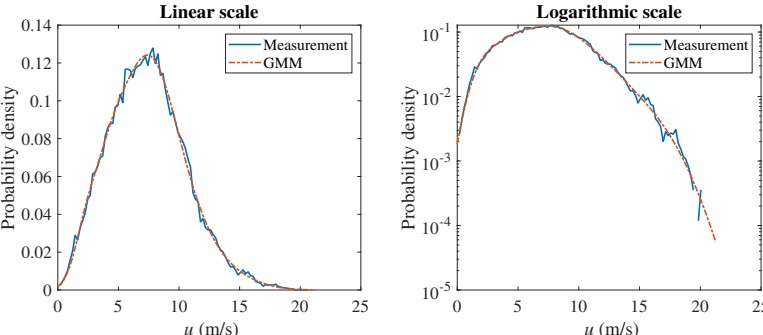

**Figure 16.** GMM marginal distribution of $u$ for onshore sector

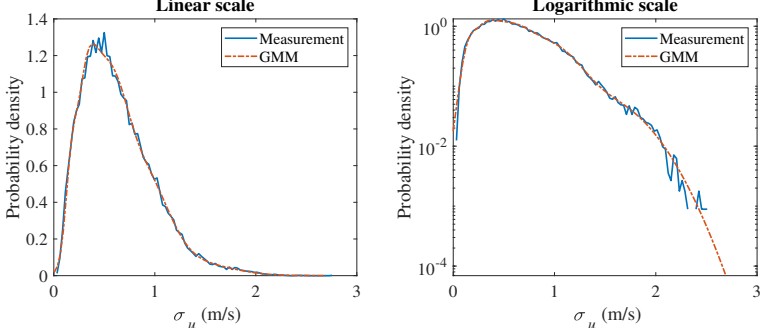

**Figure 17.** GMM marginal distribution of $\sigma_u$ for onshore sector



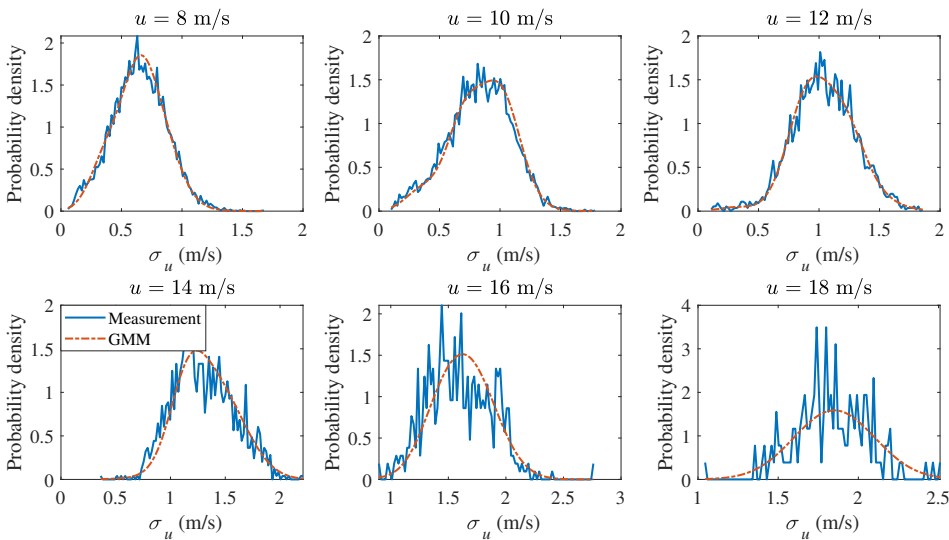

**Figure 18.** GMM probability distribution of $\sigma_u$ conditional on $u$ for onshore sector

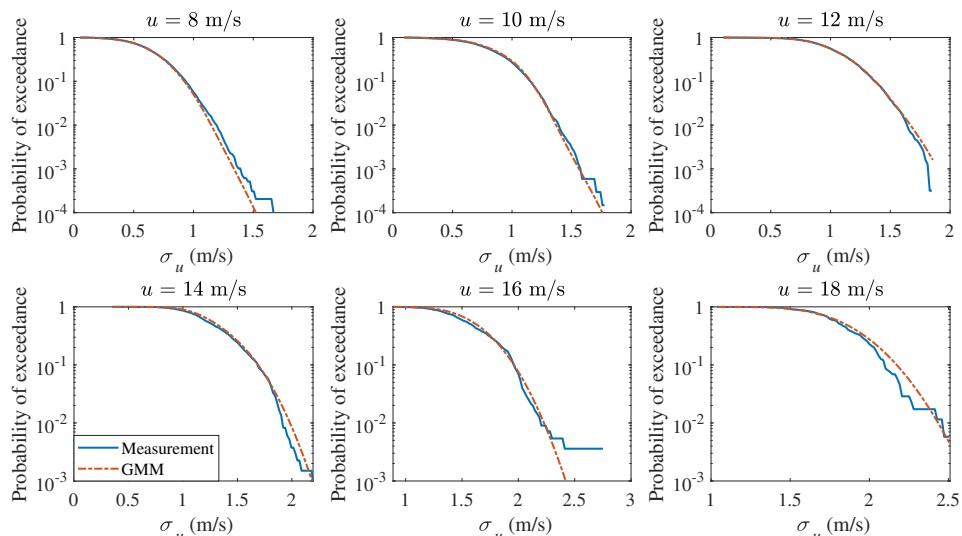

**Figure 19.** GMM probability of exceedance of $\sigma_u$ conditional on $u$ for onshore sector



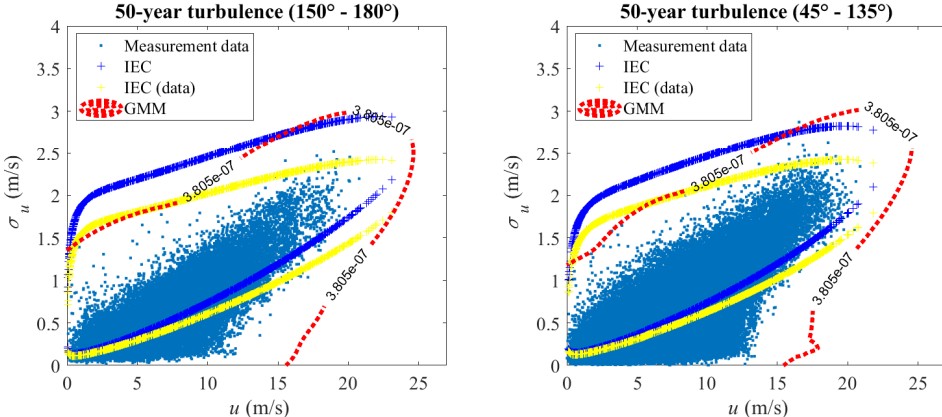

**Figure 20.** GMM and IEC 50-year turbulence estimation for two onshore sectors

## 4   Conclusions

The GMM is proposed to model the joint distribution of wind parameters, i.e., 10-minute mean wind speed and turbulence, and it is readily implementable and provides realistic 50-year turbulence levels. This model has been validated using multi-year high frequency wind velocity measurements at one site for offshore climate and for flat land terrains. Copula-based joint probability models were not found to have the flexibility to accurately model the tails of the wind turbulence distribution conditional on the mean wind speed.

A procedure using GMM that properly captures the joint distribution of wind parameters is proposed. Both the marginal distributions of mean wind speed and standard deviation, and the distribution of standard deviation conditional on mean wind speed were shown to reflect the multi-year wind measurements. This model allows a good estimation of the 50-year turbulence (validated by the marginal and conditional distributions), which serves as an input to wind turbine design load cases. The procedure of GMM is demonstrated by fitting the theoretical multivariate $t$ distribution. The GMM is then used to estimate the probability distribution of offshore wind parameters and two-sector of onshore wind parameters. There is a good agreement between the GMM estimated probability distribution and the measurement data. The 50-year turbulence is estimated from the GMM and compared with IEC. The applicability to different sectors of the wind measurement data demonstrates its flexibility and shows its potential for modelling the joint distribution of wind parameters. Compared to copula methods, it has less estimation error for the estimated marginal distributions and conditional distributions.

The determination of the optimal number of components for GMM requires further research. In this paper, four parameters is used for the theoretical $t$ distribution and eight for fitting the mean wind speed and turbulence. As more components are used, the pdf of the GMM will converge, but with more computational efforts. Another limitation for GMM is that it might not extrapolate well for certain correlation structures, especially if the sample size is small, even though the model is quite flexible.



*Author contributions.* XZ developed the methodology with contributions from AN, XZ implemented the scientific methods and validated the results, XZ wrote the original draft of the paper. AN conceived the original idea, supervised the scientific work, reviewed and edited the paper.

*Competing interests.* The authors declare that there have no competing interest.

*Acknowledgements.* This work has received funding from the Danish Energy Technology Development and Demonstration Program, EUDP,
under the project ProbWind, with grant agreement 64019-0587.



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
