# Peer review of "Gaussian mixture model for extreme wind turbulence estimation"

_Wind Energy Science, 2021_

## Referee Comment (RC1)

Overview

This paper introduces Gaussian mixture model for extreme wind turbulence estimation and applies it to a 15-year period of measurements. A discussion related to IEC standard turbulence is also included. Overall, I think the manuscript is well written and an interesting study. I have several suggestions and comments that I expect to be addressed.

Specific comments

1.      The Abstract should include the main findings of this study. You explain the research question and state that you carried out different comparative analyses between different statistical models, but it is never mentioned in the Abstract what the main finding of these analyses is. That should be added in the revised manuscript.
2.      Lines 42–43. It's common to cite papers as *Monahan (2018) model the joint…* instead of *(Monahan, 2018) model the joint…* The latter formatting is used when the references are listed to support a statement. This inconsistency is observed throughout the manuscript.
3.      Line 52: Is the provided order of 10^(-7) authors' assessment of there is a support for this value elsewhere in the literature?
4.      Line 61: Can the authors provide one more paragraph that will summarize the outcome of other studies that used GMM? Since the claim in Line 61 is that only few studies exist that used this method, it would be interesting to what were their findings and is it relevant to the present work?
5.      Line 74: The should be the (without capital t).
6.      Line 83: So, what is the value of $k$ in this study? The authors proceed with explaining the theoretical framework of this model, but the reader is not provided with the information about $k$. And how is that value of $k$ justified or determined? Update: I see now that the value is provided after Table 1, but this should be discussed even earlier so that the reader is not confused by the time it gets to Table 1.
7.      Line 86: Can you provide a reference (book or paper) for the 3-step estimation of model parameters that is described after Line 86?
8.      Section 3.1. Is there any study on homogeneity of wind data from this tower? One would assume that anemometers were re-calibrated and/or replaced and the environment around the anemometers has changed over the 15-year interval. This can introduce systematic biases to wind data. Can the authors comment on this?
9.      Lines 142, 144 and elsewhere: Use power notation for m/s and other units.
10.     Is it possible to further explore this method to separate turbulence associated with thunderstorm winds (downbursts and various gust front outflows) vs. severe non-thunderstorm winds. The former class of winds is characterized by non-Gaussian distribution of fluctuations as well as mean wind (i.e., mean wind is not constant over a 10-min period). See Hangan et al. (2019; J. Fluid Struct. Doi: 10.1016/j.jfluidstructs.2019.01.024) for further discussion.
11.     While I am not sure how to provide a sudden to improve what I am about to say, after reading this manuscript several times I have a feeling that the authors could have made better job of connecting the observational data and the proposed methodology. An example is Section 3.2., i.e., what is the main message of this section? Why randomly sampling those data when the authors later present real observations?
12.     One can perform a similar analysis to what is done in this paper by using Monte Carlo simulations where the random numbers are generated from (observed or assumed) wind distributions with the constraint that the generated numbers (fluctuations) need to obey turbulence

energy spectra. Then, one can estimate turbulence and other statistical parameters from the generated data. What are some of the positives (and perhaps negatives) of the method proposed herein in respect to the simpler Monte Carlo simulations?

13.    Line 253: How computationally expensive is this method? How much computational time was required to perform this analysis?

---

## Referee Comment (RC2)

In general, the manuscript is written with care. The math and statistics presented in the manuscript seem to be correct. However, this reviewer has the following comments

1). It is not clear throughout the manuscript what wind speed the authors are referring to. Is it the time-averaged "10-min mean wind speed" or "3-s gust mean wind speed?. The clarification throughout the manuscript. In the introduction, it is stated (Lines 35-40) "… focused on the probability distribution of wind speed standard deviation [sigma_u] conditional on the mean wind speed (u), whereas it is required that the joint distribution of [sigma_u] and u is properly modeled" Does u denote the "10-min mean wind speed" or the mean of "10-min mean wind speed"? This reviewer has a difficult time deciphering which is which. Clarification of this could significantly help this reviewer.

For clarity, in the following x10 will be used to refer to "10-min mean wind speed".

2). The term modeling "wind turbulence", "extreme wind turbulence" and "probability distribution of wind turbulence" are employed. However, the physical meaning of wind turbulence is unclear. Does it refer to x10, the standard deviation of x10, or the mean of x10?

In fact, the term "50-year turbulence levels" is not clear. Since often in wind engineering we speak 50-year return period value of annual maximum 10-min mean (or hourly mean) wind speed.

3) It is stated that "For modeling extreme turbulence accurately, the tail of the joint probability distribution of [sigma_u] and u, must be accurately represented to small exceedance probabilities of the order of $10-7$." Again, accurate representation of x10, mean of x10, or standard deviation of x10?

4) The use of GMM is interesting. However, from a Bayesian point of view, if x is normally distributed, by considering its mean and/or its standard deviation are uncertain (due to small sample size effects), its posterior distribution which is obtained as a "weighted" Gaussian distribution is still Gaussian. This aspect needs to be discussed and contrasted with the GMM considered in the submitted manuscript.

5) Line 170. It was stated k is set equal to 4. It is not clear to this reviewer why k=4 is considered.

Based on the above, this reviewer is not in the position to recommend its publication. Once the physical meaning of the terms is clearly defined, a re-review is necessary to examine the details. New queries are likely to be raised.

---

## Author Comment (AC1)

**Authors' response to reviewers' comments**

| | |
|---|---|
| Journal: | Wind Energy Science |
| Title of paper: | Gaussian mixture model for extreme wind turbulence estimation |
| Authors: | Xiaodong Zhang, Anand Natarajan |
| Manuscript No.: | wes-2021-147 |

**Authors' Response to the Comments of Reviewer #1**

The authors would like to thank the reviewer for the comments and advice on the submission. The manuscript will be revised accordingly and the detailed responses are provided below.

**Overview**: *This paper introduces Gaussian mixture model for extreme wind turbulence estimation and applies it to a 15-year period of measurements. A discussion related to IEC standard turbulence is also included. Overall, I think the manuscript is well written and an interesting study. I have several suggestions and comments that I expect to be addressed.*

**Response**: Your review of the manuscript and providing valuable comments are appreciated. The issues highlighted are addressed and changes will be made in the new submission.

**Comment 1**: *The Abstract should include the main findings of this study. You explain the research question and state that you carried out different comparative analyses between different statistical models, but it is never mentioned in the Abstract what the main finding of these analyses is. That should be added in the revised manuscript.*

**Response 1**: Thanks, the main finding is the minimal estimation error of the Gaussian mixture model to fit both the conditional distribution of the standard deviation of the 10-min wind speed and the marginal distribution of the mean of the 10-min wind speed, and this will be added to the revised manuscript.

**Comment 2**: *Lines 42–43. It's common to cite papers as Monahan (2018) model the joint... instead of (Monahan, 2018) model the joint... The latter formatting is used when the references are listed to support a statement. This inconsistency is observed throughout the manuscript.*

**Response 2**: Thanks for pointing out the difference between a narrative citation (Monahan (2018)) and a parenthetical citation (Monahan, 2018). The following in-text citation will be changed:
(Monahan, 2018) to Monahan, (2018);
(Srbinovski et al. 2021) to Srbinovski et al. (2021);
(Chang et al. 2017) to Chang et al. (2017);
(Cui et al. 2018) to Cui et al. (2018);
(Li et al. 2020) to Li et al. (2020);
and (Wahbah et al. 2018) to Wahbah et al. (2018).

**Comment 3**: *Line 52: Is the provided order of $10^{-7}$ authors' assessment of there is a support for this value elsewhere in the literature?*

**Response 3**: The value of $3.8 \times 10^{-7}$ is the probability of exceedance of 10-minute wind parameters associated with 50-year return period, which is specified in IEC 61400-1 standard. It is important to check the performance of the GMM on estimating the probability of exccedance close to the order of $10^{-7}$.

**Comment 4**: *Line 61: Can the authors provide one more paragraph that will summarize the outcome of other studies that used GMM? Since the claim in Line 61 is that only few studies exist that used this method, it would be interesting to what were their findings and is it relevant to the present work?*

**Response 4**: The authors would like to clarify that we are not claiming that few studies exist using GMM. Based on our literature review, it is found that few studies has applied GMM in wind energy industry, especially for modelling the joint distribution of wind parameters. There are many studies in other domains, e.g., speed and audio processing, image classification, density estimation of microarray data in bioinformatics, etc., as is presented in this paper. More literature review will be provided in the revised manuscript.

**Comment 5**: *Line 74: The should be the (without capital t).*

**Response 5**: Thanks for the comments, the sentence will be changed to:

"In terms of density estimation, the GMM is useful for multivariate distribution representations with multiple modes..."

**Comment 6**: *Line 83: So, what is the value of k in this study? The authors proceed with explaining the theoretical framework of this model, but the reader is not provided with the information about k. And how is that value of k justified or determined? Update: I see now that the value is provided after Table 1, but this should be discussed even earlier so that the reader is not confused by the time it gets to Table 1.*

**Response 6**: The value of $k$ represents the number of Gaussian components, which is evaluated based on data. It could be estimated by Akaike information criterion when the sample size is not too big. For this study, the $k$ is evaluated by convergence of the estimation, where $k = 4$ in the multivariate $t$ distribution example, and $k = 8$ for the wind parameter examples. More discussion on the value of $k$ will be provided earlier in the revised manuscript.

**Comment 7**: *Line 86: Can you provide a reference (book or paper) for the 3-step estimation of model parameters that is described after Line 86?*

**Response 7**: Please see Ref [1] for the k-means algorithm and Ref [2] for the Expectation-Maximization (EM) algorithm (both references are in this the Reference section of this document).

**Comment 8**: *Section 3.1. Is there any study on homogeneity of wind data from this tower? One would assume that anemometers were re-calibrated and/or replaced and the environment around the anemometers has changed over the 15-year interval. This can introduce systematic biases to wind data. Can the authors comment on this?*

**Response 8**: Please refer to [3] for more details on the data. Yes, the sensors on the Høvsøre mast have been replaced regularly and calibrated, the data used in this paper is calibrated data. This will be clarified in the revised manuscript.

**Comment 9**: *Lines 142, 144 and elsewhere: Use power notation for m/s and other units.*

**Response 9**: The unit m/s will be changed to $\mathrm{m\,s^{-1}}$ in the new submission.

**Comment 10**: *Is it possible to further explore this method to separate turbulence associated with thunderstorm winds (downbursts and various gust front outflows) vs. severe non-thunderstorm winds. The former class of winds is characterized by non-Gaussian distribution of fluctuations as well as mean wind (i.e., mean wind is not constant over a 10-min period). See Hangan et al. (2019; J. Fluid Struct. Doi: 10.1016/j.jfluidstructs.2019.01.024) for further discussion.*

**Response 10**: The objective of our article is to improve the stochastic modeling of turbulence given in the IEC 61400-1 as used in load case simulations by demonstrating an accurate conditional and marginal distribution of turbulence. The IEC 61400-1 does not consider non-stationary turbulence and accordingly in our analysis, we consider the wind speed variations to be stationary. In the future, we can widen our analysis to apply it to non-stationary wind conditions, but this would be too large a scope for the present article.

**Comment 11**: *While I am not sure how to provide a sudden to improve what I am about to say, after reading this manuscript several times I have a feeling that the authors could have made better job of connecting the observational data and the proposed methodology. An example is Section 3.2., i.e., what is the main message of this section? Why randomly sampling those data when the authors later present real observations?*

**Response 11**: The purposes of Section 3.2 are: 1) to prove that the Gaussian Mixture model could model non-Gaussian distribution with small prediction error; 2) to demonstrate the procedure of the using Gaussian Mixture model for sampling, density estimation, and tail extrapolation. To sample from the fitted joint-distribution of wind parameters is very important as many reliability analysis and uncertainty quantification applications require random samples. The revised manuscript will explain this section better.

**Comment 12**: *One can perform a similar analysis to what is done in this paper by using Monte Carlo simulations where the random numbers are generated from (observed or assumed) wind distributions with the constraint that the generated numbers (fluctuations) need to obey turbulence energy spectra. Then, one can estimate turbulence and other statistical parameters from the generated data. What are some of the positives (and perhaps negatives) of the method proposed herein in respect to the simpler Monte Carlo simulations?*

**Response 12**: Monte-Carlo simulations require an accurate joint-distribution of turbulence and wind speed to be utilized. If this is already at hand, then one can add the constraint that the random value for turbulence is satisfying the corresponding spectra. This article is focusing on the former aspect, that is obtaining an accurate joint-distribution of turbulence and wind speed, because this has not been accomplished in the present design standards and load simulations conducted thereof. Without an accurate joint-distribution of turbulence and wind speed, the constraint for the random turbulence value to match the spectral energy by itself cannot provide an accurate probability of occurrence of that turbulence value.

**Comment 13**: *Line 253: How computationally expensive is this method? How much computational time was required to perform this analysis?*

**Response 13**: The procedure could be done within several minutes on a standard laptop and is dependent on the sample size.

**Authors' Response to the Comments of Reviewer #2**

The authors would like to thank the reviewer for the comments and advice on the submission. The manuscript will be revised accordingly and the detailed responses are provided below.

**Overview**: *In general, the manuscript is written with care. The math and statistics presented in the manuscript seem to be correct. However, this reviewer has the following comments*

**Response**: Your review of the manuscript and providing valuable comments are appreciated. The issues highlighted are addressed and changes will be made in the new submission.

**Comment 1**: *It is not clear throughout the manuscript what wind speed the authors are referring to. Is it the time averaged "10-min mean wind speed" or "3-s gust mean wind speed?. The clarification throughout the manuscript. In the introduction, it is stated (Lines 35-40) "... focused on the probability distribution of wind speed standard deviation $\sigma_u$ conditional on the mean wind speed (u), whereas it is required that the joint distribution of $\sigma_u$ and u is properly modeled". Does u denote the "10-min mean wind speed" or the mean of "10-min mean wind speed"? This reviewer has a difficult time deciphering which is which. Clarification of this could significantly help this reviewer. For clarity, in the following ×10 will be used to refer to "10-min mean wind speed"*

**Response 1**: $u$ denotes mean of the longitudinal wind speed over a 10-minute time duration. This will be clarified and consistent in the revised manuscript.

**Comment 2**: *The term modeling "wind turbulence", "extreme wind turbulence" and "probability distribution of wind turbulence" are employed. However, the physical meaning of wind turbulence is unclear. Does it refer to x10, the standard deviation of x10, or the mean of x10? In fact, the term "50-year turbulence levels" is not clear. Since often in wind engineering we speak 50-year return period value of annual maximum 10-min mean (or hourly mean) wind speed.*

**Response 2**: "wind turbulence" refers to the standard deviation of the longitudinal wind speed over a 10-minute time duration. This will also be clarified and consistent in the revised manuscript.

**Comment 3**: *It is stated that "For modeling extreme turbulence accurately, the tail of the joint probability distribution of $\sigma_u$ and u, must be accurately represented to small exceedance probabilities of the order of $10^{-7}$." Again, accurate representation of 10, mean of ×10, or standard deviation of ×10?*

**Response 3**: Accurate representation of the joint distribution of $\sigma_u$ (standard deviation of 10-min wind speed) and $u$ (mean of 10-min wind speed). This will be clarified and consistent in the revised manuscript.

**Comment 4**: *The use of GMM is interesting. However, from a Bayesian point of view, if x is normally distributed, by considering its mean and/or its standard deviation are uncertain (due to small sample size effects), its posterior distribution which is obtained as a "weighted" Gaussian distribution is still Gaussian. This aspect needs to be discussed and contrasted with the GMM considered in the submitted manuscript.*

**Response 4**: Thanks for the comment. To evaluate the variance error of the prediction due to small sample size has not been investigated in this study and will be our future work. The GMM is estimated based on sample size that is larger than $10^4$ for all the examples in this study. Even though a weighted sum of Gaussian random variables is a Gaussian random variable, a weighted Gaussian distribution

is not necessarily Gaussian. When there are more than two components for the GMM, the GMM is multi-modal and is not Gaussian distributed. This will be discussed in the revised manuscript.

**Comment 5**: *Line 170. It was stated k is set equal to 4. It is not clear to this reviewer why k=4 is considered.*

**Response 5**: Thanks for the comments. The $k = 4$ is selected by the convergence study. Starting from $k = 1$, a GMM is fitted, and $k$ will be increased to $k+1$ to fit the GMM with one more component, the convergence of the fitted GMMs will be checked. The $k$ stops increasing when the fitted distribution is converged. Following this procedure, $k = 4$ for this example. The Akaike information criterion (AIC) could be used for selecting the value of $k$ when the sample size is relatively small, but with large sample size (e.g., $> 10^4$), it requires further study to find a proper criterion for determining the value of $k$, which is described in line 251 of our submitted manuscript. The choice of $k$ value in will be discussed in more detail in the revised manuscript.

**Comment 6**: *Based on the above, this reviewer is not in the position to recommend its publication. Once the physical meaning of the terms is clearly defined, a re-review is necessary to examine the details. New queries are likely to be raised.*

**Response 6**: Thanks again for the comments.

**References**

[1] D. Arthur and S. Vassilvitskii, "K-means++: The Advantages of Careful Seeding," p. 9.

[2] G. J. McLachlan, S. X. Lee, and S. I. Rathnayake, "Finite Mixture Models," p. 26, 2019.

[3] A. Peña, R. Floors, A. Sathe, S.-E. Gryning, R. Wagner, M. S. Courtney, X. G. Larsén, A. N. Hahmann, and C. B. Hasager, "Ten Years of Boundary-Layer and Wind-Power Meteorology at Høvsøre, Denmark," *Boundary-Layer Meteorology*, vol. 158, pp. 1–26, Jan. 2016.

---

## Author Response (AR1)

**Authors' response to reviewers' comments**

| | |
|---|---|
| Journal: | Wind Energy Science |
| Title of paper: | Gaussian mixture model for extreme wind turbulence estimation |
| Authors: | Xiaodong Zhang, Anand Natarajan |
| Manuscript No.: | wes-2021-147 |

**Authors' Response to the Comments of Reviewer #1**

The authors would like to thank the reviewer for the comments and advice on the submission. The manuscript has be revised accordingly and the detailed responses are provided below.

**Overview**: *This paper introduces Gaussian mixture model for extreme wind turbulence estimation and applies it to a 15-year period of measurements. A discussion related to IEC standard turbulence is also included. Overall, I think the manuscript is well written and an interesting study. I have several suggestions and comments that I expect to be addressed.*

**Response**: Your review of the manuscript and providing valuable comments are appreciated. The issues highlighted are addressed and changes have be made in the revised manuscript.

**Comment 1**: *The Abstract should include the main findings of this study. You explain the research question and state that you carried out different comparative analyses between different statistical models, but it is never mentioned in the Abstract what the main finding of these analyses is. That should be added in the revised manuscript.*

**Response 1**: Thanks, the main finding is the minimal estimation error of the Gaussian mixture model to fit both the conditional distribution of the standard deviation of the 10-min wind speed and the marginal distribution of the mean of the 10-min wind speed, and the following sentence is added to the end of Abstract in the revised manuscript.

**Changes 1**: In line 19:"The Gaussian mixture model is able to model the joint distribution of wind parameters well, where the estimated tail distributions of both the marginal distributions and conditional distribution have good accuracy, and it is a good candidate for extreme turbulence estimation."

**Comment 2**: *Lines 42–43. It's common to cite papers as Monahan (2018) model the joint. . . instead of (Monahan, 2018) model the joint. . . The latter formatting is used when the references are listed to support a statement. This inconsistency is observed throughout the manuscript.*

**Response 2**: Thanks for pointing out the difference between a narrative citation "Monahan (2018))" and a parenthetical citation "(Monahan, 2018)". The following in-text citations have be changed:

**Changes 2**: (Monahan, 2018) to Monahan (2018);

(Srbinovski et al. 2021) to Srbinovski et al. (2021);

(Chang et al. 2017) to Chang et al. (2017);

(Cui et al. 2018) to Cui et al. (2018);

(Li et al. 2020) to Li et al. (2020);

and (Wahbah et al. 2018) to Wahbah et al. (2018).

**Comment 3**: *Line 52: Is the provided order of $10^{-7}$ authors' assessment of there is a support for this value elsewhere in the literature?*

**Response 3**: The value of $3.8 \times 10^{-7}$ is the probability of exceedance of loads/responses within 10-min time duration associated with 50-year return period, which is specified in IEC 61400-1 standard. To avoid confusion, the sentence has been changed to:

**Changes 3**: In line 57: "To model the extreme turbulence well, both the main body and the tail of the joint probability distribution of $\sigma_u$ and $u$ should be accurately represented."

**Comment 4**: *Line 61: Can the authors provide one more paragraph that will summarize the outcome of other studies that used GMM? Since the claim in Line 61 is that only few studies exist that used this method, it would be interesting to what were their findings and is it relevant to the present work?*

**Response 4**: The authors would like to clarify that we are not claiming that few studies exist using GMM. Based on our literature review, it is found that few studies has applied GMM in wind energy industry, especially for modelling the joint distribution of wind parameters. There are many studies in other domains, e.g., speed and audio processing, image classification, density estimation of microarray data in bioinformatics, etc., as is presented in this paper. More literature review has been be provided and the paragraph has be rewritten as:

**Changes 4**: In line 57: "To model the extreme turbulence well, both the main body and the tail of the joint probability distribution of $\sigma_u$ and $u$ should be accurately represented. Gaussian mixture model (GMM) is broadly used for clustering tasks (Zhang et al., 2021). GMM is a flexible model which can also perform density estimation on multivariate data with different marginal distributions and correlation structures. It is widely applied to different fields of study, e.g., speech and audio processing (Reynolds and Rose, 1995), image classification (Permuter et al., 2003), density estimation of microarray data in bioinformatics (Steinhoff et al., 2003), cancer classification (Prabakaran et al., 2019) and finance (Miyazaki et al., 2014). GMM is less commonly applied in wind energy compared to other domains, Chang et al. (2017) used GMM based neural network for short-term wind power forecast, Cui et al. (2018) used GMM for fitting the probability distribution of wind power ramping features, Zhang et al. (2019) used GMM for wind turbine power dispatching, Li et al. (2020) used GMM for electrical loads forecast, and Srbinovski et al. (2021) used GMM for modelling the site-specific wind turbine power curves. GMM has been

rarely adopted for wind parameters modelling, Wahbah et al. (2018) used univariate GMM for wind speed probability density

55   estimation, where the joint distribution of wind speed with other parameters was not investigated. Few published literature uses GMM for density estimation of wind inflow parameters and GMM has not been used for modelling the joint distribution of $u$ and $\sigma_u$. "

**Comment 5**: *Line 74: The should be the (without capital t).*

**Response 5**: Thanks for the comments, the sentence has been changed to:

60   **Changes 5**: In line 81: "In terms of density estimation, the GMM is useful for multivariate distribution representations with multiple modes..."

**Comment 6**: *Line 83: So, what is the value of k in this study? The authors proceed with explaining the theoretical framework of this model, but the reader is not provided with the information about k. And how is that value of k justified or determined? Update: I see now that the value is provided after Table 1, but this should be discussed even earlier so that the reader is not*

65   *confused by the time it gets to Table 1.*

**Response 6**: The value of $k$ represents the number of Gaussian components, which is evaluated based on data. It could be estimated by Akaike information criterion when the sample size is not too big. For this study, the $k$ is evaluated by convergence of the estimation, where $k = 4$ in the multivariate $t$ distribution example, and $k = 8$ for the wind parameter examples. More discussion on the value of $k$ will be provided earlier in the revised manuscript.

70   **Changes 6**: In line 93 :"Some information criteria are proposed in the literature (Akaike, 1998; Schwarz, 1978) to determine $k$, where $k$ is selected as a balance of overfitting and underfitting. Nevertheless, when the sample size is too large, the criteria are not effective and further research is required."

In line 127:"To compute the number of components $k$, it is increased from 1 until the estimated density function converges."

In line 142:"The estimated density function converges when the number of components $k = 4$,..."

75   In line 240:"Note that the estimated density function converges when $k = 8$ for all the joint distribution estimations of wind parameters using GMM."

**Comment 7**: *Line 86: Can you provide a reference (book or paper) for the 3-step estimation of model parameters that is described after Line 86?*

**Response 7**: Please see (Arthur and Vassilvitskii, 2006) for the k-means algorithm and (McLachlan et al., 2019) for the

80   Expectation-Maximization (EM) algorithm (both references are in this the Reference section of this document).

**Changes 7**: In line 96: "The initial model parameters are calculated from the clusters evaluated by the $k$-means clustering algorithm (Arthur and Vassilvitskii, 2006), and optimized by the Expectation-Maximization (EM) algorithm (McLachlan et al., 2019) as follows:".

**Comment 8**: *Section 3.1. Is there any study on homogeneity of wind data from this tower? One would assume that anemometers were re-calibrated and/or replaced and the environment around the anemometers has changed over the 15-year interval. This can introduce systematic biases to wind data. Can the authors comment on this?*

**Response 8**: Please refer to Peña et al. (2016) for more details on the data. Yes, the sensors on the Høvsøre mast have been replaced regularly and calibrated, the data used in this paper is calibrated data. The following sentence has been added:

**Changes 8**: In line 157:"The sensors on the Høvsøre mast have been replaced regularly and calibrated, the data used in this paper is calibrated data (Peña et al., 2016)."

**Comment 9**: *Lines 142, 144 and elsewhere: Use power notation for m/s and other units.*

**Response 9**: The unit m/s has been changed to $\mathrm{m\,s^{-1}}$.

**Changes 9**: Please see line 176, 178, 210 and 217 for the changes.

**Comment 10**: *Is it possible to further explore this method to separate turbulence associated with thunderstorm winds (downbursts and various gust front outflows) vs. severe non-thunderstorm winds. The former class of winds is characterized by non-Gaussian distribution of fluctuations as well as mean wind (i.e., mean wind is not constant over a 10-min period). See Hangan et al. (2019; J. Fluid Struct. Doi: 10.1016/j.jfluidstructs.2019.01.024) for further discussion.*

**Response 10**: The objective of our article is to improve the stochastic modeling of turbulence given in the IEC 61400-1 as used in load case simulations by demonstrating an accurate conditional and marginal distribution of turbulence. The IEC 61400-1 does not consider non-stationary turbulence and accordingly in our analysis, we consider the wind speed variations to be stationary. In the future, we can widen our analysis to apply it to non-stationary wind conditions, but this would be too large a scope for the present article.

**Changes 10**: In line 159:"The wind speed variation is considered to be stationary, and non-stationary wind conditions are not included in this study."

**Comment 11**: *While I am not sure how to provide a sudden to improve what I am about to say, after reading this manuscript several times I have a feeling that the authors could have made better job of connecting the observational data and the proposed methodology. An example is Section 3.2., i.e., what is the main message of this section? Why randomly sampling those data when the authors later present real observations?*

**Response 11**: The purposes of Section 3.2 are: 1) to prove that the Gaussian Mixture model could model non-Gaussian distribution with small prediction error; 2) to demonstrate the procedure of the using Gaussian Mixture model for sampling, density estimation, and tail extrapolation. To sample from the fitted joint-distribution of wind parameters is very important as many reliability analysis and uncertainty quantification applications require random samples. The revised manuscript will explain this section better.

**Changes 11**: In line 122:"GMM is proposed to model the joint distribution of $u$ and $\sigma_u$, where the estimation error is small at both the main body pdf and the tail distribution. To verify the use of GMM, it is firstly used to recover the multivariate $t$ distribution from a $t$ distribution random sample. The flexibility of GMM (especially for modelling non-Gaussian joint distribution) and the demonstration of the procedure of using GMM for density estimation is detailed. To sample from the fitted joint-distribution is very important as many reliability analysis and uncertainty quantification applications require random samples as inputs. The random samples from GMM are compared with the random sample from the $t$ distribution and wind parameters. To compute the number of components $k$, it is increased from 1 until the estimated density function converges.

Using copulas to develop non-Gaussian joint distributions of the $u$ and $\sigma_u$ is initially attempted. A joint probability distribution of the $u$ and $\sigma_u$ is then modelled by GMM. For estimating the extreme turbulence (wind parameter contour with 50-year return period), the accuracy of tail distribution is important. The probability of exceedance of $\sigma_u$ conditional on $u$ from GMM is thus compared with the measurement data. To further examine the flexibility of GMM, the wind measurement data from both the offshore and onshore sectors are investigated and the 50-year wind parameter contour are compared."

Besides, section 3.1 and section 3.2 are swapped for a better flow of writing.

**Comment 12**: *One can perform a similar analysis to what is done in this paper by using Monte Carlo simulations where the random numbers are generated from (observed or assumed) wind distributions with the constraint that the generated numbers (fluctuations) need to obey turbulence energy spectra. Then, one can estimate turbulence and other statistical parameters from the generated data. What are some of the positives (and perhaps negatives) of the method proposed herein in respect to the simpler Monte Carlo simulations?*

**Response 12**: Monte-Carlo simulations require an accurate joint-distribution of turbulence and wind speed to be utilized. If this is already at hand, then one can add the constraint that the random value for turbulence is satisfying the corresponding spectra. This article is focusing on the former aspect, that is obtaining an accurate joint-distribution of turbulence and wind speed, because this has not been accomplished in the present design standards and load simulations conducted thereof. Without an accurate joint-distribution of turbulence and wind speed, the constraint for the random turbulence value to match the spectral energy by itself cannot provide an accurate probability of occurrence of that turbulence value.

**Comment 13**: *Line 253: How computationally expensive is this method? How much computational time was required to perform this analysis?*

**Response 13**: The procedure could be done within several minutes on a standard laptop and is dependent on the sample size.
**Changes 13**: In line 260:"...several minutes on a standard laptop computer..."

The authors would like to thank the reviewer for the comments and advice on the submission. The manuscript will be revised
145    accordingly and the detailed responses are provided below.

**Overview**: *In general, the manuscript is written with care. The math and statistics presented in the manuscript seem to be correct. However, this reviewer has the following comments*

**Response**: Your review of the manuscript and providing valuable comments are appreciated. The issues highlighted are addressed and changes will be made in the new submission.

150    **Comment 1**: *It is not clear throughout the manuscript what wind speed the authors are referring to. Is it the time averaged "10-min mean wind speed" or "3-s gust mean wind speed?. The clarification throughout the manuscript. In the introduction, it is stated (Lines 35-40) "... focused on the probability distribution of wind speed standard deviation $\sigma_u$ conditional on the mean wind speed (u), whereas it is required that the joint distribution of $\sigma_u$ and u is properly modeled". Does u denote the "10-min mean wind speed" or the mean of "10-min mean wind speed"? This reviewer has a difficult time deciphering which*
155    *is which. Clarification of this could significantly help this reviewer. For clarity, in the following $\times 10$ will be used to refer to "10-min mean wind speed"*

**Response 1**: $u$ denotes mean of the longitudinal wind speed over a 10-minute time duration. This has been clarified and consistent in the revised manuscript. The following changes have been made:

**Changes 1**: In line 4:" The wind parameters (mean and standard deviation of longitudinal wind speed over 10-min time
160    duration)...";

In line 12:" In this paper, the Gaussian mixture model is used to model the joint distribution of mean and standard deviation of longitudinal wind speed over 10-min time duration ..."

In line 42:"literature have mainly focused on the probability distribution of the standard deviation of the wind speed ($\sigma_u$) conditional on the mean of the longitudinal wind speed over a 10-minute time duration ($u$), whereas it is required that the joint
165    distribution of $\sigma_u$ and $u$ is properly modeled."

In line 155:"$u$ and $\sigma_u$, which is linearly detrended..."

In line 229:" The $u$ and $\sigma_u$ in the onshore section (150° to 180°) are..."

Changes are also made in line 239, 244, 247, 249, and 260.

**Comment 2**: *The term modeling "wind turbulence", "extreme wind turbulence" and "probability distribution of wind turbu-*
170    *lence" are employed. However, the physical meaning of wind turbulence is unclear. Does it refer to x10, the standard deviation*

*of x10, or the mean of x10? In fact, the term "50-year turbulence levels" is not clear. Since often in wind engineering we speak 50-year return period value of annual maximum 10-min mean (or hourly mean) wind speed.*

**Response 2**: "wind turbulence" is described by the standard deviation of the longitudinal wind speed over a 10-minute duration. This will also be clarified and consistent in the revised manuscript.

**Changes 2**: In line 3: "wind turbulence (described by the standard deviation of the longitudinal wind speed over a 10-minute time duration) has a significant impact..."

**Comment 3**: *It is stated that "For modeling extreme turbulence accurately, the tail of the joint probability distribution of $\sigma_u$ and u, must be accurately represented to small exceedance probabilities of the order of $10^{-7}$." Again, accurate representation of 10, mean of $\times 10$, or standard deviation of $\times 10$?*

**Response 3**: Accurate representation of the joint distribution of $\sigma_u$ (standard deviation of 10-min wind speed) and $u$ (mean of 10-min wind speed). This will be clarified and consistent in the revised manuscript.

**Changes 3**: In line 57: "To model the extreme turbulence well, both the main body and the tail of the joint probability distribution of $\sigma_u$ and $u$ should be accurately represented."

**Comment 4**: *The use of GMM is interesting. However, from a Bayesian point of view, if x is normally distributed, by considering its mean and/or its standard deviation are uncertain (due to small sample size effects), its posterior distribution which is obtained as a "weighted" Gaussian distribution is still Gaussian. This aspect needs to be discussed and contrasted with the GMM considered in the submitted manuscript.*

**Response 4**: Thanks for the comment. To evaluate the variance error of the prediction due to small sample size has not been investigated in this study and will be our future work. The GMM is estimated based on sample size that is larger than $10^4$ for all the examples in this study. Even though a weighted sum of Gaussian random variables is a Gaussian random variable, a weighted Gaussian distribution is not necessarily Gaussian. When there are more than two components for the GMM, the GMM is multi-modal and is not Gaussian distributed. This will be discussed in the revised manuscript.

**Changes 4**: In line 83:"GMM is a linear combination of multivariate Gaussian distribution components, where each component is defined by its mean and covariance. Even though a weighted sum of Gaussian random variables is a Gaussian random variable, a weighted Gaussian distribution is not necessarily Gaussian. When there are more than two components for GMM, the GMM is multi-modal and is not Gaussian distributed."

**Comment 5**: *Line 170. It was stated k is set equal to 4. It is not clear to this reviewer why k=4 is considered.*

**Response 5**: Thanks for the comments. The $k = 4$ is selected by the convergence study. Starting from $k = 1$, a GMM is fitted, and $k$ will be increased to $k+1$ to fit the GMM with one more component, the convergence of the fitted GMMs will be checked. The $k$ stops increasing when the fitted distribution is converged. Following this procedure, $k = 4$ for this example. The Akaike information criterion (AIC) could be used for selecting the value of $k$ when the sample size is relatively small, but with large

sample size (e.g., $> 10^4$), it requires further study to find a proper criterion for determining the value of $k$, which is described in line 251 of our submitted manuscript. The choice of $k$ value in will be discussed in more detail in the revised manuscript.

[revised manuscript text omitted]